# Care of Adult Heart Transplant Recipients by the Primary Care Provider: A Practical Roadmap

**DOI:** 10.3390/jcm14041346

**Published:** 2025-02-18

**Authors:** Yu Wu, Lina Gao, Jose Lazo, Brandon Martinez, Simran Grewal, Irina Yurkova, Julia Galeota, Catherine Nash, Meghan Cutler, Nimaljeet Tarango, Pooja Prasad, Richard Cheng, Shweta Motiwala, Teresa De Marco

**Affiliations:** 1Department of Surgery, Division of Cardiothoracic Surgery, Section of Mechanical Circulatory Support, University of California San Francisco Health, San Francisco, CA 94143, USA; 2Department of Medicine, Division of Cardiology, Section of Advanced Heart Failure and Heart Transplant, University of California San Francisco Health, San Francisco, CA 94143, USAmeghan.cutler@ucsf.edu (M.C.);; 3Department of Pharmacy, University of California San Francisco Health, San Francisco, CA 94143, USA; jose.lazo@ucsf.edu (J.L.);; 4Department of Social Work, University of California San Francisco Health, San Francisco, CA 94143, USA

**Keywords:** heart transplantation, front-line healthcare provider, immunosuppression, infections, psychosocial care

## Abstract

Heart transplantation has significantly improved survival and enhanced the quality of life of patients with end-stage heart failure. Successful long-term outcomes are predicated on a collaborative effort among patients, transplant teams, and primary care providers (PCPs). Notably, PCPs are increasingly pivotal in post-transplant care, engaging in annual assessments, early recognition of complications, and referral, thus minimizing morbidity and mortality. This article highlights key considerations for PCPs, including indications for heart transplant, immunosuppressive therapy and infection prophylaxis, management of post-transplant complications, psychosocial and lifestyle adjustment, and family planning. This roadmap aims to empower PCPs to deliver optimal care and improve long-term outcomes for heart transplant recipients.

## 1. Introduction

Orthotopic heart transplantation (OHT), first successfully performed in 1967 [1], has transformed the management of end-stage or advanced heart failure (AHF), significantly improving survival rates and quality of life for over 100,000 recipients in the United States [2]. Thanks to technological advancements and expansion of the donor pool, the number of OHTs performed annually continues to rise, having nearly doubled over the past two decades [2]. In 2024, 4572 OHTs were performed in the United States [2].

The survival rate for OHT recipients remains optimal, with a 1-year survival of 91% and a median survival of 12–13 years [3]. The long-term success of OHT hinges on multidisciplinary care. Primary care providers (PCPs) play an important role in the prevention, detection, and management of both short-term and long-term complications inherent in immunocompromised transplant recipients. This review highlights key aspects in the care of adult OHT recipients, including indications for OHT, immunosuppressive (IS) therapy and infection prophylaxis, management of post-transplant complications, psychosocial and lifestyle adjustment, and family planning.

### 1.1. What Are the Indications for a Heart Transplant?

OHT should be considered for patients with AHF who have a limited life expectancy and/or poor quality of life, particularly when the following treatment options have either proven ineffective or are deemed inappropriate: (1) severe functional limitations classified as NYHA III or IV despite optimal medical therapy [4]; (2) severe or recurrent myocardial ischemia not amenable to revascularization [5,6]; (3) recurrent/refractory dysrhythmias that significantly increases the risk of sudden death or results in an unacceptable quality of life due to frequent implantable cardioverter defibrillator (ICD) discharges [5,6]; (4) other conditions that place the patient at risk of sudden death or decompensation.

### 1.2. How Is Heart Transplantation Performed?

In the biatrial technique, both the superior and inferior vena cava (SVC and IVC) are connected to the right atrium (RA) and left atrium (LA). Typically, the SVC connects to the RA, while the IVC connects to the LA. This older technique was often used in patients with congenital heart conditions, including single-ventricle physiology. It allows systemic venous return to reach the lungs for oxygenation without needing a functioning right ventricle.

In contrast, bicaval anastomosis, the modern technique, connects both the SVC and IVC directly to the RA and bypasses the LA. The technique provides better hemodynamic stability by maintaining the function and geometry of RA.

Caveats:

-The biatrial technique creates two sinus nodes, which results in the appearance of two P-waves in the electrocardiogram (ECG).
oThe donor atrial activity is conducted within the ventricle, generating the QRS complex.oNative atrial activity does not cross the suture line.

### 1.3. What Are the Physiologic Changes After Heart Transplantation?

Surgical denervation following OHT leads to an increased resting heart rate (HR) (90–130 bpm) [7] and chronotropic incompetence due to sympathetic and parasympathetic sectioning [8]. This results in a blunted HR response to exercise, reduced maximal HR, and delayed HR recovery to baseline after exercise [8], all of which are influenced by circulating catecholamines primarily from peripheral nerves [9]. Notably, partial sympathetic reinnervation of the sinus node, which can be observed 1 year after transplant, has been demonstrated to improve HR responses and exercise capacity [9].

Considerations:

-Medications that primarily or completely act through the autonomic nervous system are ineffective.oAnticholinergic agents (atropine), which are vagotonic agents, do not elicit elevation of HR in the setting of parasympathetic denervation.oα2-agonists (Clonidine) are unable to lower HR and blood pressure (BP) due to the loss of sympathetic innervation to the heart including the vagus nerve and central sympathetic output.oDigoxin has a vagolytic effect and in the setting of parasympathetic denervation, it fails to act as an AV nodal blocker; however, its direct action on NA/K ATPase remains effective in increasing contractility.Direct-acting agents such as beta blockers (BBs), calcium channel blockers (CCBs), and beta-adrenergic agents remain effective.

## 2. Medications

### 2.1. What Medications Do OHT Recipients Need to Take?

After transplantation, a recipient’s immune system is intentionally suppressed to reduce the risk of rejection. They are considered immunocompromised due to the need for lifelong immunotherapy. Maintenance immunosuppressants (ISs) not only prevent rejection but also protect against cardiac allograft vasculopathy (CAV) while minimizing the risk of infection and malignancy through the use of the lowest effective dose [10]. Standard regimens include tacrolimus, mycophenolate, and corticosteroids [11]. The introduction of tacrolimus, in place of cyclosporine, has improved the median graft survival to 11.3 years [12]. Mammalian targets of rapamycin inhibitors (mTORIs), such as sirolimus and everolimus, may help reduce CAV, although the indication and timing for mTORIs introduction can vary widely [13,14,15,16]. Azathioprine has largely been replaced by mycophenolate for superior survival outcomes [17]. Patient education on the importance of adherence to ISs is vital, and the adverse effects of ISs and significant drug–drug interactions (Table 1) must be monitored.

While ISs are essential for preventing organ rejection in OHT patients, the use of ISs comes with several risks and complications. These include the risk of infections, cancers, kidney injury, hypertension, hyperglycemia, and osteoporosis. These complications will be discussed in the section dedicated to the care of post-transplant complications.

### 2.2. What Types of Vaccinations Are Required Before and After OHT?

Pre-transplant vaccination should be completed at least 2 weeks before transplantation for inactivated vaccines and 4 weeks for live vaccines (Table 2). Incomplete post-transplant vaccines should prompt a consultation with transplant and infectious disease specialists [18,19]. Most centers restart vaccinations 3–6 months post-transplant, but defer them during any period of active rejection treatment. Vaccination status should be reviewed before traveling to high-risk areas, ensuring both routine and travel-specific vaccines are considered.

Role of PCPs:

-Identify potential opportunities to update vaccinations.-Administer inactive vaccinations within the community (Table 2).

### 2.3. What Medications Can Be Used for Treating COVID-19 in OHT Recipients?

Managing COVID-19 in transplant recipients in an outpatient setting can be challenging due to the interactions between ISs and COVID-19 medications. Remdesivir is preferred for its favorable safety profile. In contrast, Nirmatrelvir/Ritonavir (Paxlovid) should be used with caution due to its significant interaction with CNIs and mTORIs, which could lead to toxicity [20]. It is crucial to monitor patients closely and consult with transplant specialists when considering Nirmatrelvir/Ritonavir (Paxlovid). A dose reduction or temporary withholding of ISs may be recommended.

Principles of care for PCPs:

-Before initiating any drugs for the treatment of COVID-19, explore potential interactions and discuss with transplant specialists.

## 3. Complications

### 3.1. What Are Commonly Seen Complications After OHT?

#### 3.1.1. Rejection

Lifelong IS treatment is required to prevent allograft rejection. However, rejection remains one of the major complications post-transplant, significantly affecting graft and patient survival [21]. Routine clinical visits and monitoring with surveillance tests are essential for early diagnosis and treatment. The tests include endomyocardial biopsy, gene expression profiling, and tests for donor-derived cell-free DNA and donor-specific antibodies. Additionally, immune cell function is also monitored frequently post-transplant [21]. These tests are commonly performed and interpreted by heart transplant cardiologists. 

Rejection may be asymptomatic, underscoring the need for regular monitoring [21]. Patients with suspected rejection should be immediately referred to their transplant center. Endomyocardial biopsy remains the gold standard for diagnosis, while transthoracic echocardiogram (TTE) is utilized to assess cardiac function [21]. The treatment of rejection depends on the type and severity of rejection.

Principles of care for PCPs:

-Patients with new symptoms (fever, chills, fatigue, dyspnea, relative hypotension, arrhythmias, edema, and weight gain) should be urgently referred to their primary transplant center.-In the case of delays, PCPs should maintain close communication with the transplant center and may initiate empiric treatments including an oral prednisone bolus dose and a tapered regimen as an outpatient or IV methylprednisolone [21] if there is any evidence of allograft dysfunction.

#### 3.1.2. Cardiac Allograft Vasculopathy

CAV, characterized by diffuse intimal thickening of coronary arteries, is a leading cause of graft failure [21]. Its prevalence escalates from 8% at 1 year to 50% at 10 years post-transplant [21]. Due to cardiac denervation, patients typically do not present with angina [22], which makes it challenging to diagnose early. Coronary angiography is recommended as a diagnostic test [21]. Preventive strategies include the use of statins and mTORIs, while treatment options such as percutaneous coronary intervention can be considered in selected patients and re-transplantation in advanced cases.

Principles of care for PCPs:

-Refer patients promptly to their primary transplant centers if ischemia is suspected. Patients may present with ventricular arrhythmias, decompensated HF, or anginal symptoms (although angina is rare, the symptoms may involve persistent crushing, chest pain, and dyspnea) [22].-If a patient presents to an emergency room, it is crucial to perform an ECG promptly to detect any dysrhythmias. Additionally, cardiac enzymes should be obtained with close consultation with the primary transplant center.

#### 3.1.3. Dyslipidemia

Dyslipidemia affects up to 71% of OHT recipients due to IS therapy, hypertension, diabetes, and obesity [23]. It significantly increases the risk of CAV, the leading cause of death 2–4 years post-transplant [23]. Regular lipid monitoring and statin therapy are recommended for all patients, regardless of cholesterol level, starting at low doses due to potential interactions with CNIs [23].

Principles of care for PCPs:

-CNIs increase the risk of statin-induced rhabdomyolysis. Monitor for muscle pain and check creatine kinase if symptoms occur, and statin discontinuation should be entertained.-Initiate lower-than-usual doses of preferred agents like pravastatin, rosuvastatin, and atorvastatin.-Alternative agents are Ezetimibe, PSK9 inhibitors, and bempedoic acid.-For hypertriglyceridemia, consider using omega 3 alpha ethyl esters, eicosapent ethyl, and fenofibrate.-Cholestyramine and colestipol impede CNI absorption and must be administered cautiously in post-transplant patients.

#### 3.1.4. Infection

Infection, a leading cause of post-transplant mortality, accounts for 34.3%, 14.3%, and 12.2% of deaths at 1 month–1 year, 1–3 years and 3–5 years, respectively [24]. Prophylaxis and prompt management are essential for optimizing post-heart transplant outcomes, such as vaccinations, prophylactic measures, and individualized treatments. ISHLT guidelines recommend tailored strategies for pathogens such as cytomegalovirus (CMV), Pneumocystis jiroveci pneumonia (PJP), Aspergillus, Coccidioides, and Toxoplasma gondii, which should be based on the serologies of donors and recipients (Table 3) [25].

Principles of care for PCPs:

-Any medication adjustment should be discussed with transplant specialists to ensure the continuity and efficacy of ISs.-Review the drug–drug interactions of any new drugs being added to the regimen of OHT recipients and discuss with primary transplant specialists.

Role of PCPs:

-Bactrim is prescribed to prevent PJP and toxoplasmosis; any plans to discontinue this medication should prompt a consultation with transplant specialists.-Consider consulting transplant cardiologists, infectious disease specialists, and pharmacists when treating OHT recipients with Hepatitis C and/or human immunodeficiency virus.-Any immunocompromised patient presenting with fever (even low grade) and nonspecific symptoms (fatigue/malaise, chills, cough, dyspnea, sore throat, headache, nausea, vomiting or diarrhea, confusion or altered mental status, urinary symptoms, incision pain or redness) should raise concerns about infection and prompt immediate evaluation and consultation with transplant specialists. The evaluations include the following:oCBC, procalcitonin, blood and urine cultures, CMV polymerase chain reaction, and chest x-ray (CXR)oStool cultures, ova, and parasites, as well as C. difficile for any patient with diarrheal illness


#### 3.1.5. Hypertension

Hypertension occurs in 50–95% of recipients [21], primarily due to CNI use and corticosteroid [26], as well as associated comorbidities such as chronic kidney disease (CKD) or diabetes. Aggressive BP control (<120 mmHg systolic) is suggested to improve outcomes, especially for those with CAV [26]. Routine monitoring of BP and appropriate management, including lifestyle modifications and medication adjustments, are essential for effective BP control [21].

Principles of care for PCPs:

-Acceptable antihypertensive therapy includes the following:oAngiotensin-converting enzyme inhibitors or angiotensin receptor blockers can be considered.Caveat: hyperkalemia and renal insufficiency are a concern in the setting of CNI.oDihydropyridine CCB (DHP-CCBs) such as amlodipine and nifedipine. Caveat: Non-DHP CCBs such as diltiazem and verapamil should be used cautiously given their interaction with CNIs resulting in high CNI levels which could worsen renal function.oOther agents such as vasodilators (Hydralazine), and alpha 2 agonists (clonidine) can also be consideredoBBs, especially combined beta and alpha blockers, can be considered for BP management in OHT recipients, but bradycardia may limit their utility. In the setting of tachycardia, these agents are useful for managing hypertension.

#### 3.1.6. Diabetes

Approximately 20% of OHT recipients develop steroid-induced diabetes within the first year after their transplant, and 23% of patients will have diabetes after 5 years [27,28]. Routine monitoring for diabetes after OHT should include checking fasting plasma glucose, hemoglobin A1c, and fructose amine in select patients [28]. The management of new-onset or preexisting diabetes typically involves a combination of insulin therapy and oral anti-glycemic medications. Although data are sparse for post-transplant patients, sodium–glucose transport protein 2 (SGLT2) inhibitors and glucagon-like peptide 1 agonists (GLP-1) can also be considered for their nephroprotective benefits [28,29].

Principles of care for PCPs:

-Monitor closely for steroid-induced hyperglycemia when a patient remains on steroid post-transplant.-Metformin should be considered in patients with adequate renal function, with instruction to hold off if a patient is acutely ill, at risk for hypovolemia, or scheduled for a procedure.-The use of SGLT2 and GLP-1 should be carried out in consultation with transplant cardiologists, endocrinologists, and nephrologists.-SGLT2 and GLP-1 are recommended for OHT recipients with CV risks [27,28].-Monitor glucose closely and adjust anti-glycemic agents as needed while using GLP-1 [28].

Caveats:

-SGLT2 may increase the risk of mycotic urinary tract infections.-GLP-1 agonists should be avoided for patients with type 1 diabetes.-The use of dapsone may falsely lower a patient’s A1c; therefore, fructose amine monitoring should be substituted to assess the effectiveness of therapy.

#### 3.1.7. Chronic Kidney Disease

CKD is a progressive, long-term complication in OHT recipients, affecting 1.9%, 10.9%, and 21% of patients at 1, 5 and 10 years post-transplant, respectively [21,30]. CKD can be caused by acute tubular necrosis, CNI, mTORI, and other drug toxicities. Fluctuating CNI levels in the immediate post-transplant period increase the risk of ne-phrotoxicity [30]. This significantly impacts patient morbidity and mortality, and often requires advanced interventions like dialysis or kidney transplant [21]. Therefore, it is essential to closely monitor renal function and medication adjustments based on an individual’s clinical scenario to mitigate the risk of kidney injury.

Principles care for PCPs:

-Monitor renal function and urine protein creatinine ratio every 3 months in patients on mTORIs.-Avoid nephrotoxins such as NSAIDs and contrast dye unless necessary.-Collaborate with the primary heart transplant team to minimize CNI exposure.

#### 3.1.8. Dysrhythmias

Atrial and ventricular tachycardia affect about 10–25% of OHT recipients. This is primarily due to the complete denervation of the donor’s heart and the loss of vagal tone [31]. Bradycardia happens in 18–27% of transplanted hearts immediately, being associated with factors such as preservation injury, compromised blood supply to sinoatrial node, or drug-induced [31]. In the majority of cases, bradycardia is usually resolved within 3 weeks [32]. The incidence of pacemaker implant post-transplant is reported to be approximately 1.5% [32,33], with consideration of implantation if bradycardia persists for more than 3 weeks.

Role of PCPs:

-If new atrial or ventricular dysrhythmia is detected, evaluation tests (ECG, TTE, cardiac enzymes) should be initiated to rule out allograft rejection, in close consultation with a primary transplant center.-If a patient’s hemodynamics are stable, consider using Type 1 agents (i.e., procainamide) and amiodarone to treat atrial dysrhythmias (SVT or atrial flutter/fibrillation) and ventricular dysrhythmias (ventricular flutter/fibrillation).-Consider consulting a primary transplant center to initiate CCB and BB use to treat atrial dysrhythmias. Of note, non-DHP CCBs have a significant drug-drug interaction with immunosuppressants.-If necessary, only a low dose of peripheral IV Adenosine 3–6 mg should be considered as adenosine receptors can be upregulated and this may result in significant bradycardia or a high-grade block.-If the patient is hemodynamically unstable, refer to the local emergency room and transfer to the primary transplant center for further management.

#### 3.1.9. Malignancy

Malignancy remains a leading cause of death 5–10 years post-transplant [34], with risk factors including pre-transplant malignancy, older recipient age, male sex, and white race [34]. Skin cancer is the most common, followed by prostate and lung cancers. The 2023 ISHLT guidelines recommend regular skin cancer surveillance including education on preventive measures (e.g., wearing sunscreen, a hat, and clothes with long sleeves) and yearly dermatological exams [21]. Additionally, staying up to date with mammograms, pap smears, prostate-specific antigen testing, oral checks with dentists, and colonoscopies are also important for the early detection of malignancy.

Role of PCPs:

-Ensure that patients are up to date with all recommended screenings.-Coordinate local tests and share results with the transplant center.

#### 3.1.10. Osteoporosis

The systemic use of glucocorticoids and CNIs can lead to increased bone loss, muscle atrophy, and higher adiposity, contributing to osteoporosis that affects up to 57% of post-transplant recipients [21]. Bisphosphonates, in combination with calcium and vitamin D, are prescribed for both prevention and treatment.

Role of PCPs:

-Monitor bone health with dual-energy X-ray absorptiometry scans.-Collaborate with endocrinologists and transplant specialists to develop a treatment plan for osteoporosis [21].

## 4. Lifestyle Adjustment

### 4.1. What Kind of Lifestyle Adjustments Do OHT Recipients Experience?

Nutrition: After OHT, it is important for recipients to maintain a heart-healthy diet, which includes limiting the intake of salt and saturated fats [35]. The consumption of unpasteurized dairy products and raw or undercooked meats should be avoided to reduce the risk of infection [35]. Additionally, grapefruit should be excluded from the diet due to its significant interaction with CNIs, which are commonly used in post-transplant IS therapy.

Exercise: Rehabilitation begins during hospitalization and transitions to outpatient care, incorporating lifelong exercise for optimal recovery. Exercise programs should include warm-up, cool-down, and aerobic exercises to improve VO_2_ max and resistance training to improve function and bone density [8]. Initial moderate-intensity continuous training is recommended and then tailored high-intensity interval training is introduced as early as 11 weeks post-surgery [9].

Principles of care for PCPs:

-Sternal precautions are required for the first 6–8 weeks post-transplant to prevent sternal nonunion.-Cardiac output primarily depends on SV increases via the Frank–Starling mechanism, not HR.-It is important to include warm-up and cool-down sessions in the rehabilitation exercise program due to the features of cardiac denervation.

Daily life and work: After transplant, recipients have frequent clinic visits and tests, including cardiac biopsies during the first year. However, the frequency of these visits typically decreases thereafter. Generally, OHT recipients are able to return to work 6–12 months post-transplant, especially those with good recovery and who are engaged in non-manual labor [36,37]. Additionally, recipients may resume driving approximately 8–12 weeks after transplant, depending on their recovery progress [37]. Notably, studies have indicated that individuals who return to work post-transplantation have a lower mortality risk [36].

### 4.2. What Are the Psychosocial Concerns Post-Transplant for Patients and Their Caregivers?

OHT recipients face various challenges, including lifestyle adjustments, emotional turmoil from the risk of rejection, adherence to IS treatment, and uncertainties about long-term survival [21,38]. These challenges, along with concerns about body image, financial strain, and family difficulties, can increase the risk of anxiety and depression, which are linked to higher mortality rates [38,39,40]. Studies have showed that a comprehensive transplant team and strong family support are predictors of positive outcomes in both physical and psychosocial health-related quality of life at 1–5 years post-transplant [41]; patients with robust emotional and social support tend to experience fewer depressive symptoms in the long term, maintain higher self-esteem, and report greater life satisfaction [41,42]. Therefore, identifying caregivers and establishing a support network pre-transplant is vital for successful recovery and the promotion of well-being. 

Role of PCPs:

-Collaborate with the transplant center to ensure patients receive appropriate community resources and mental health assistance as needed.

## 5. Family Planning

### 5.1. Is It Possible to Get Pregnant After a Transplant?

Yes. With improved survival and quality of life, pregnancy is increasingly considered by young, childbearing-aged women following a transplant. The ISHLT recommends waiting at least 8 weeks after transplant before resuming sexual activity [21]. It also advises against pregnancy in the first 12 months after a transplant due to the higher risk of rejection and complications and the teratogenic effects of ISs [21,43]. Preconception counseling is essential for achieving favorable outcomes and should involve careful family planning with multidisciplinary teams. Pregnancy may be viable for stable patients who have had no recent episodes of rejection, infection, or HF, but it is discouraged in those with graft dysfunction and significant CAV [21,43,44,45].

#### Principles of Care for PCPs

Preconception:

-Hormonal contraception (combined estrogen/progestin, progestin alone) may be considered, and intrauterine devices are recommended.

Postpartum:

-Intrauterine devices are preferred; avoid combined hormonal contraception for transplant recipients with active liver disease and CAV.-A progesterone subdermal implant is an acceptable method of long-term contraception-Adjustment of anti-hypertensives may be required for patients using estrogen- containing contraceptives.-Maternal cardiac function, renal, and hepatic function should be monitored periodically throughout and after pregnancy. This should include appropriate imaging and laboratory tests.-Collaborate with both the transplant center and obstetrics and gynecology teams to coordinate maternity care.-Mycophenolate use should be avoided due to teratogenic risk.

## 6. Conclusions

Long-term success in post-transplant care is a collaborative effort among patients, transplant teams, and PCPs. PCPs are playing an increasingly more important role in this process. Table 4 summarizes the clinical pearls for PCPs while caring for OHT recipients. Effective collaboration between PCPs and transplant specialists, along with early detection and management of complications, is vital in optimizing patient outcomes and improving the quality of life of OHT recipients.

## Figures and Tables

**Table 1 jcm-14-01346-t001:** Maintenance Immunosuppression for Heart Transplant Recipients.

Medication	Clinical Pearls	Common Adverse Effects	Notable Drug Interactions
Tacrolimus (Prograf, Envarsus, Astagraf)	Therapeutic drug monitoring routinely performed; extended-release product is not interchangeable with immediate release products 12 h trough level	Neurological: tremors and headaches Renal: nephrotoxicity, hyperkalemia, hypomagnesemia Tacrolimus: alopecia;, post-transplant diabetes	3A4 inhibitors: Macrolide antibiotics, Diltiazem, Azole antifungals 3a4 inducers: Rifampin, Nafcillin, Carbamazepine, Phenytoin
Cyclosporine (Sandimmune, Neoral, Gengraf)	Therapeutic drug monitoring routinely performed; non-modified and modified cyclosporine products are not interchangeable 12 h trough level	Same as tacrolimus Cyclosporine: hirsutism and gingival hyperplasia	Same as tacrolimus
Mycophenolate mofetil (Cellcept) Mycophenolate acid (Myfortic)	REMS program regarding pregnancy prevention and planning, myfortic may decrease frequency of GI effects	Gastrointestinal: diarrhea, abdominal pain, nausea Hematologic: leukopenia and thrombocytopenia	Aluminum/magnesium hydroxide (decreases concentration of mycophenolate) Agents that enhance myelosuppression
mTORI: Sirolimus (Rapamune), Everolimus (Zortress)	Therapeutic drug monitoring routinely performed, data to support decreased occurrence of CAV 12 h trough	Delayed wound healing, mouth ulcers, peripheral edema, proteinuria, hypertriglyceridemia	Same as tacrolimus

Abbreviations: mTORI, mammalian target of rapamycin inhibitor; CAV, coronary allograft vasculopathy.

**Table 2 jcm-14-01346-t002:** Vaccinations for transplant recipients.

Vaccine	Recommended Before Transplant	Recommended After Transplant
Influenza	Yes	Yes
COVID	Yes	Yes
Hepatitis B	Yes	Yes
Hepatitis A	Yes	Yes
Tetanus	Yes	Yes
Pertussis (Tdap)	Yes	Yes
Inactivated Polio	Yes	Yes
H influenza type B	Yes	Yes
S pneumonia (conjugate/polysaccharide)	Yes	Yes
Rabies	Yes	Yes
Human papilloma virus	Yes	Yes
Varicella (live attenuated)	Yes	No
Varicella (subunit)	Yes	Yes
Measles/Mumps/Rubella	Yes	No
BCG	Yes	No
Smallpox	No	No
Anthrax	No	No

Adapted from Danzige-Isakov et al. [18].

**Table 3 jcm-14-01346-t003:** Infection management recommendations.

Infection	Pathogen	Prophylaxis	Treatment	Notes
Viral	CMV	PO valganciclovir	IV ganciclovir Foscarnet or Cidofovir Maribavir for treatment-refractory cases	Based on donor/recipient CMV serologic status, monitor for leukopenia and neutropenia. Monitor CMV viral load and perform genotypic resistance testing for refractory cases.
	COVID	NA	Nirmatrelvir/Ritonavir (Paxlovid) Remdesivir	Contact transplant team if considering Nirmatrelvir/Ritonavir (Paxlovid) for transplant recipients due to significant interaction risks with calcineurin and mTOR inhibitors
Fungal	Aspergillus	Fluconazole Posaconazole	Posaconazole, Voriconazole Isavuconazole, and lipid formulations of amphotericin B as alternatives	Monitor for drug interactions and hepatic enzyme elevation. Voriconazole carries a higher risk of skin cancer that may want to be avoid in transplant population if alternatives are available. Diagnosis relies on serum and bronchoalveolar lavage galactomannan assays, and Beta-D-Glucan tests.
	Pneumocystis Jiroveci	Trimethoprim-sulfamethoxazole (TMP-SMX) Atovaquone Dapsone- check G6PD	1st line: TMP-SMX 2nd line: IV pentamidine	highest risk within first six months post-transplant. Adjunctive corticosteroids recommended in hypoxemic patients.
	Candida	Fluconazole and or Itraconazole	1st line: Azoles (Fluconazole) Echinocandins (Caspofungin, micafungin) 2nd line: Amphotericin B	Consider previous exposures and potential drug interactions.
	Coccidioides	NA	Fluconazole or Itraconazole for treatment; Amphotericin B for severe cases	6–12 months for recipients in endemic areas or from donors with prior active coccidioidomycosis.
Protozoal	Toxoplasma gondii	TMP-SMX with additional pyrimethamine for seronegative recipients from seropositive donors (D+/R−)	TMP-SMX	Lifelong prophylaxis may be necessary for high-risk recipients. Some centers recommend an additional six weeks of pyrimethamine.

Adapted from Sparkes et al. [25]. Abbreviations: CMV, Cytomegalovirus; TMP-SMX, trimethoprim-sulfamethoxazole. NA, not applicable.

**Table 4 jcm-14-01346-t004:** Clinical pearls of post heart transplant care for primary care providers.

Stages of Post OHT	Main Concerns	Questions to Patients	Tests	Recommendations
Early <12 months	- Rejection - Infection - Acute kidney injury - Vaccinations - HTN - Diabetes Mellitus/Hyperglycemia - Dyslipidemia	- Have you been experiencing any HF symptoms? (i.e., shortness of breath, trouble lying flat, leg swelling, weight gain, or palpitations) since last visit? - How has your blood pressure been? - How have your blood sugars been? - Have you had any recent sick contacts? - Have you developed any fever/chills or diarrhea? - How is your mood? Perform psychological screening	- Rejection: TTE, EKG - Routine labs: CBC w/differential, BNP, CMP, fasting glucose, HbA1C, li-pid profile. - Infection Workup: CXR, stool studies (C.diff, ova and parasites, bacterial stool culture), UA with reflex to culture, blood culture, RVP, COVID RNA CPR, CMV PCR, procalcitonin	If there are concerns regarding progressive heart failure symptoms, rejection, or active infection, initiate workup and refer to heart transplant specialists immediately. - Acute Kidney Injury: evaluate volume status and current diuretic use. Refer to transplant center for concerns. - Anti-HTN: goal SBP < 140 mmHg - Provide vaccinations based on recommendations (see Table 2). - Hyperglycemia: antidiabetic medications and insulin as needed; consult endocrinology. - Physical Function: cardiac rehabilitation referral. - Dyslipidemia: low dose statin regardless of lipid profile. Consider up-titration if therapy is inadequate. PCSK9 inhibitors can be safely and effectively used in statin intolerance or in refractory hyperlipidemia. - Refer for psychological counseling and therapy as needed for emotional and psychiatric symptoms. Referral to a heart transplant program social worker can be made. May require pharmacologic therapy in some cases; consult with the heart transplant center before initiation.
Middle 1–3 years	- HTN - Hyperglycemia - Dyslipidemia - CKD - Osteoporosis - Physical Function	- How has your blood pressure been? - How have your blood sugars been? - Have you had any difficulty with urination (urgency, frequency, difficulty initiating, etc.). - Are you able to carry out daily activities? - Have you completed cardiac rehab?	- Vital signs - Age specific cancer screening - Routine labs: CBC w/differential, BNP, CMP, fasting glucose, HbA1C, lipid profile. - Bone Density Monitoring	- AntiHTN: same as above. - Hyperglycemia: same as above. - Dyslipidemia: same as above. - CKD: Consider initiation of SGLT2i if appropriate, consider referral to Nephrology. - Physical Function: same as above. - Osteoporosis—Referral to endocrinology. Evaluate for vitamin D deficiency and hypocalcemia.
Late >3 years –5	- CAV - Malignancy - CKD - HTN - Hyperglycemia - Dyslipidemia - CKD - Osteoporosis - HTN - Psychosocial concerns	- Have you experienced any chest pain, shortness of breath, fatigue, or fluid retention since your last visit? Have you completed your annual transplant testing (stress test and ECHO)? - Have you had any difficulty with urination (urgency, frequency, difficulty initiating, etc.) - Have you visited your dermatologist and ophthalmologist this year? - Have you been experiencing any symptoms of anxiety or depression?	- ECG - Age specific cancer screening: mammogram, prostate exam, colonoscopy, etc., - Routine labs: CBC w/differential, BNP, CMP, fasting glucose, HbA1C, lipid profile. Bone Density Monitoring - Anxiety and Depression screening	- For concerns regarding CAV or heart failure symptoms, refer to heart transplant specialists immediately - Malignancy: Annual surveillance testing by dermatology and ophthalmology, dentist, in addition to age specific cancer screening. Oncology referral if diagnosis confirmed or suspected. - CKD: same as above - Osteoporosis—Referral to endocrinology. Evaluate for vitamin D deficiency and hypocalcemia. - Anti-HTN: same as above - Anxiety/Depression: Referral to psychologist/psychiatry as needed

Abbreviations: OHT, orthotopic heart transplantation; TTE, transthoracic echocardiograph; ECG, electrocardiogram; CXR, chest X-ray; CAV, cardiac allograft vasculopathy; HTN, hypertension; CKD, chronic kidney disease, CBC, complete blood count; BNP, brain natriuretic peptide; CMP, complete metabolic panel; RVP, respiratory viral.

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
