# Peer review of "Care of Adult Heart Transplant Recipients by the Primary Care Provider: A Practical Roadmap"

_jcm, 2025, doi:10.3390/jcm14041346_

Round 1
Reviewer 1 Report
Comments and Suggestions for Authors
Abstract:
The abstract is well-organized but could benefit from slightly more fluid transitions between the listed topics. For instance, consider rephrasing the list to improve readability, such as: "This article highlights key considerations for PCPs, including: indications for heart transplantation, immunosuppressive therapy and infection prophylaxis, management of post-transplant complications, and psychosocial and lifestyle adjustments."
The abstract does a good job of outlining the topics covered.However, consider briefly emphasizing the growing importance of PCPs in managing heart transplant recipients to underscore the paper's relevance.
Replace "late-stage heart failure" with "end-stage heart failure" for greater alignment with standard clinical terminology.
Add a closing sentence emphasizing the practical utility of the article for PCPs, such as: "This roadmap aims to empower PCPs to deliver optimal care and improve long-term outcomes for heart transplant recipients."
Introduction:
The introduction effectively provides a detailed historical context for orthotopic heart transplantation (OHT), its clinical significance, and advancements in surgical techniques. It sets a strong foundation for the paper.
Some sentences are overly long and complex. Breaking them into shorter, more concise statements would improve readability, particularly for readers unfamiliar with the subject. For example, “Long-term success in OHT hinges on multidisciplinary care…” could be simplified and split into two sentences.
Minor errors such as "condcuted" instead of "conducted" and "sever" instead of "severe" should be corrected. Proofreading is recommended to ensure the text is error-free. (Professional Proofreading is highly recommended)
While the importance of PCPs is mentioned, providing a brief example or context of their unique contributions (e.g., early detection of complications, managing immunosuppression side effects) would enhance the narrative. The survival statistics and annual OHT numbers are informative but could be presented in a more engaging way, such as comparing historical and current data to emphasize progress over time.
Ensure consistency in the use of font type and size throughout the document to maintain a professional and cohesive appearance. This refinement will enhance readability and align the presentation with academic and publication standards. (Ex: Line 79-83)
General Comments:
The article lacks a conventional structure typically expected in a narrative review. To enhance clarity and coherence, it is strongly recommended to restructure the article into the following standard sections:
Introduction: This section should provide a succinct overview of the topic, outline the objectives of the review, and establish its significance, particularly the role of primary care providers (PCPs) in managing heart transplant recipients.
Methodology: A methodology section is essential and currently missing. It should detail the process of selecting and reviewing the literature, including the databases searched, inclusion and exclusion criteria, and any framework used for evaluating the quality of the studies. This addition is crucial to ensure transparency and credibility of the review.
Discussion: A discussion section is necessary to synthesize the reviewed information, highlight key findings, and connect them to broader implications. This section should also address gaps in the literature, limitations of the review, and potential directions for future research.
Conclusion: Conclude the review with a concise summary of the key takeaways and the practical implications for PCPs. Emphasize the importance of their role in improving outcomes for heart transplant recipients.
Author Response
1. The abstract is well-organized but could benefit from slightly more fluid transitions between the listed topics. For instance, consider rephrasing the list to improve readability, such as: "This article highlights key considerations for PCPs, including indications for heart transplantation, immunosuppressive therapy and infection prophylaxis, management of post-transplant complications, and psychosocial and lifestyle adjustments."
Thanks for your comment. We have revised the sentence “This article highlights key considerations for PCPs, including indications for heart transplant, immunosuppressive therapy and infection prophylaxis, management of post-transplant complications, psychosocial and lifestyle adjustment, and family planning.”
2. The abstract does a good job of outlining the topics covered. However, consider briefly emphasizing the growing importance of PCPs in managing heart transplant recipients to underscore the paper's relevance.
Thanks for your comment. We have added “Notably, PCPs are increasingly pivotal in post-transplant care, engaging in annual assessments, early recognition of complications, and referral, thus minimizing morbidity and mortality.”
3. Replace "late-stage heart failure" with "end-stage heart failure" for greater alignment with standard clinical terminology.
Thanks for your comment. We have revised to “end-stage heart failure”.
4. Add a closing sentence emphasizing the practical utility of the article for PCPs, such as: "This roadmap aims to empower PCPs to deliver optimal care and improve long-term outcomes for heart transplant recipients."
Thanks for your comment. We have added "This roadmap aims to empower PCPs to deliver optimal care and improve long-term outcomes for heart transplant recipients."
5. Introduction:
The introduction effectively provides a detailed historical context for orthotopic heart transplantation (OHT), its clinical significance, and advancements in surgical techniques. It sets a strong foundation for the paper.
Some sentences are overly long and complex. Breaking them into shorter, more concise statements would improve readability, particularly for readers unfamiliar with the subject. For example, “Long-term success in OHT hinges on multidisciplinary care…” could be simplified and split into two sentences.
Thanks for your comments and suggestions. We have rephrased the sentence to “Long-term success in OHT hinges on multidisciplinary care. Primary care providers (PCPs) play an important role in the prevention, detection, and management of both short-term and long-term complications inherent in immunocompromised transplant recipients.”
6. Minor errors such as "condcuted" instead of "conducted" and "sever" instead of "severe" should be corrected. Proofreading is recommended to ensure the text is error-free. (Professional Proofreading is highly recommended)
Thanks for your suggestion. Errors have been corrected. Professional Proofreading is utilized in the manuscript revision.
7. While the importance of PCPs is mentioned, providing a brief example or context of their unique contributions (e.g., early detection of complications, managing immunosuppression side effects) would enhance the narrative. The survival statistics and annual OHT numbers are informative but could be presented in a more engaging way, such as comparing historical and current data to emphasize progress over time.
Thanks for your comments and suggestion. We agreed with your recommendation. We have added “Thanks to technological advancements and expansion of the donor pool, the number of OHTs performed annually continues to rise, nearly doubled over the past two decades.² In 2024, 4,572 OHTs were performed in the United States.²”
8. Ensure consistency in the use of font type and size throughout the document to maintain a professional and cohesive appearance. This refinement will enhance readability and align the presentation with academic and publication standards. (Ex: Line 79-83)
Thanks for your comment. The use of Arial and size 12 is throughout the manuscript.
9. General Comments:
The article lacks a conventional structure typically expected in a narrative review. To enhance clarity and coherence, it is strongly recommended to restructure the article into the following standard sections:
Introduction: This section should provide a succinct overview of the topic, outline the objectives of the review, and establish its significance, particularly the role of primary care providers (PCPs) in managing heart transplant recipients.
Methodology: A methodology section is essential and currently missing. It should detail the process of selecting and reviewing the literature, including the databases searched, inclusion and exclusion criteria, and any framework used for evaluating the quality of the studies. This addition is crucial to ensure transparency and credibility of the review.
Discussion: A discussion section is necessary to synthesize the reviewed information, highlight key findings, and connect them to broader implications. This section should also address gaps in the literature, limitations of the review, and potential directions for future research.
Conclusion: Conclude the review with a concise summary of the key takeaways and the practical implications for PCPs. Emphasize the importance of their role in improving outcomes for heart transplant recipients.
Thanks for your comments and suggestion.
Systematic review and meta-analysis are much needed in such a topic. However, the format of systematic review is out of the scope of our review manuscript. Furthermore, we followed the format for review manuscript by JCM. Please see the recently JCM published review example entitled “Exercise Training in Heart Failure: Current Evidence and Future Directions” https://doi.org/10.3390/jcm14020359
Additionally, it was also the format utilized by Dr. J. Kobashigawa in his review article entitled “Management of Heart Transplant Recipients: Reference for Primary Care Physicians” https://doi.org/10.3810/pgm.2012.07.2563 and Dr. S. Sehgal in his review article entitled “Heart Transplant in Children: What a Primary Care Provider Needs to Know”. https://doi.org/10.3928/19382359-20180319-01
Again, we appreciated your kind suggestion. We will consider using systematic review method for our next manuscript.
Reviewer 2 Report
Comments and Suggestions for Authors
The article summarises in a useful and practical way several critical aspects in the management of heart transplant recipients in a primary care setting. The form is that of a kind of narrative review divided into areas with proposed summaries of the evidence and descriptions of the key points for the PCP. pertaining to the journal.
I have a few observations
line 43 chap. 1 please explain the relevance with the aim
line 52 chap. 2 does not seem relevant with the aim
line 46 page 7 please detail further empiric treatment
line 57 page 8 anginal symptoms but previously it is stated that they are rare, please define furtherly signs and symptoms of suspicion and the role of ECG monitoring in primary care settings
line 60 page 12 add the reference
8.7 and 8.8 probably to be united
it might be useful to add a specific understanding for nutritional, rehabilitation/ daily life, work, mental health aspects, with practical considerations as for the other chapters. in addition, the relationship of PCPs with patient associations and citizen education.
line 144 page 14 to be revised with sexual life, contraception and pregnancy for instances.
Author Response
Comments and Suggestions for Authors
1. The article summarises in a useful and practical way several critical aspects in the management of heart transplant recipients in a primary care setting. The form is that of a kind of narrative review divided into areas with proposed summaries of the evidence and descriptions of the key points for the PCP. pertaining to the journal.
Thank you for your kind words and support.
I have a few observations
2. line 43 chap. 1 please explain the relevance with the aim
Thanks for your comment. Because the target audience is primary care providers, the paragraph serves as a brief explanation why heart transplant is needed.
3. line 52 chap. 2 does not seem relevant with the aim
Thanks for your comment. Majority of primary care providers do not know how heart transplant surgery is performed nor the anatomical changes with heart transplant surgery. It serves as fundamental knowledge to explain physiologic changes after transplant including ECG changes, blunted HR response to exercise, and different response to various medications.
Therefore, we illustrated the surgical techniques briefly to fulfill the knowledge gap with the anatomical changes with heart transplant surgery.
4. line 46 page 7 please detail further empiric treatment
Thanks for your comment and suggestion. We agreed and added the explanation of empiric treatment “initiate empiric treatment including an oral prednisone bolus dose and a tapered regimen as an outpatient or IV methylprednisolone”
5. line 57 page 8 anginal symptoms but previously it is stated that they are rare, please define further signs and symptoms of suspicion and the role of ECG monitoring in primary care settings
Thanks for your comments and suggestions. We agreed and added the signs and symptoms and ECG utilization in cases with anginal symptoms. The paragraphs have been revised as following:
- Refer patients promptly to their primary transplant centers if ischemia is suspected. Patients may present with ventricular arrhythmias, decompensated HF, or anginal symptoms (although angina is rare, the symptoms may involve persistent crushing, chest pain, and dyspnea). 22
- If a patient presents to an emergency room, it is crucial to perform an ECG promptly to detect any dysrhythmias. Additionally, cardiac enzymes should be obtained with close consultation with the primary transplant center.
6. line 60 page 12 add the reference
Thanks for your suggestion. The reference has been added “Sammour Y, Nassif M, Magwire M, et al. Effects of GLP-1 receptor agonists and SGLT-2 inhibitors in heart transplant patients with type 2 diabetes: Initial report from a cardiometabolic center of excellence. JHLT. 2021;40(6).4426-429. https://doi.org/10.1016/j.healun.2021.02.012”
9. 8.7 and 8.8 probably to be united
Thanks for your suggestion. We combined these two sections as following:
CKD is a progressive, long-term complication in OHT recipients, accounting for 1.9%, 10.9%, and 21% at 1, 5 and 10 years, respectively.21,30 CKD can be caused by nephrotoxicity, which may arise due to acute tubular necrosis, CNI, mTORI, and other drug toxicities. Fluctuating CNI levels in the immediate post-transplant period increases the risk of nephrotoxicity.30 It significantly impacts patient morbidity and mortality, and often requires advanced interventions like dialysis or kidney transplant.21 Therefore, it is essential to closely monitor renal function and medication adjustments based on an individual’s clinical scenarios to mitigate the risk of kidney injury.
Principles care for PCPs:
- Monitor renal function and urine protein creatinine ratio every 3 months in patients on mTORI.
- Avoid nephrotoxins: NSAIDs and contrast dye unless necessary.
- Collaborate with the primary heart transplant team to minimize CNI exposure.
10. It might be useful to add a specific understanding for nutritional, rehabilitation/ daily life, work, mental health aspects, with practical considerations as for the other chapters. in addition, the relationship of PCPs with patient associations and citizen education.
Thanks for your comments and suggestions. We have added the new section regarding lifestyle adjustment, addressing nutrition, rehab.
See the paragraph below:
Lifestyle adjustment
What kind of lifestyle adjustment OHT recipients experience?
Nutrition: after OHT, it is important for recipients to maintain a heart-healthy diet, which includes limiting the intake of salt and saturated fats. 35 Consumption of unpasteurized dairy products and raw or undercooked meats should be avoided to reduce the risk of infection. 35 Additionally, grapefruit should be excluded from the diet due to its significant interaction with CNI, which is commonly used in post-transplant IS therapy.
Exercise: Rehabilitation begins during hospitalization and transitions to outpatient care, incorporating lifelong exercise for optimal recovery. Exercise programs should include warm-up, cool-down, aerobic exercise to improve VOâ‚‚ max, and resistance training to improve function and bone density.8 Initial moderate-intensity continuous training is recommended and then tailored high-intensity interval training is introduced as early as 11 weeks post-surgery.9
Principles of care for PCPs:
- Sternal precaution is required for the first 6-8 weeks post-transplant to prevent sternal nonunion.
- Cardiac output primarily depends on SV increase via Frank-Starling mechanism, not HR.
- It’s important to include warm-up and cool-down sessions in the rehabilitation exercise program due to the cardiac denervation feature.
Daily life and work: After transplant, recipients have frequent clinic visits and tests, including cardiac biopsies during the first year. However, the frequency of these visits typically decreases thereafter. Generally, OHT recipients are able to return to work 6-12 months post-transplant, especially those with great recovery and engaged in non-manual labor.36,37 Additionally, recipients may resume driving approximately 8-12 weeks after transplant, depending on their recovery progress. 37 Notably, studies have indicated that individuals who return to work post-transplantation have a lower mortality risk. 36
Aspect of work and mental health have been addressed in the section of psychosocial concerns.
11. line 144 page 14 to be revised with sexual life, contraception and pregnancy for instances.
Thanks for your comment. We have revised the title to “Family planning”
Is it possible to get pregnant after transplant?
Yes. With improved survival and quality of life, pregnancy is increasingly considered for young, childbearing-aged women following a transplant. The ISHLT recommends waiting at least 8 weeks after transplant before resuming sexual activity. 21 It also advises against pregnancy in the first 12 months after transplant due to the higher risk of rejection and complications, and teratogenic effects from IS.21,43 Preconception counseling is essential for achieving favorable outcomes and should involve careful family planning with multidisciplinary teams. Pregnancy may be viable for stable patients who have had no recent episodes of rejection, infection, or HF, but it is discouraged in those with graft dysfunction and significant CAV.21,43-45
Round 2
Reviewer 1 Report
Comments and Suggestions for Authors
The response to the review comments are adequate.